# Posterior Network: Uncertainty Estimation without OOD Samples via Density-Based Pseudo-Counts

**Bertrand Charpentier, Daniel Zügner, Stephan Günnemann**
Technical University of Munich, Germany
{charpent, zuegnerd, guennemann}@in.tum.de

## Abstract

Accurate estimation of aleatoric and epistemic uncertainty is crucial to build safe and reliable systems. Traditional approaches, such as dropout and ensemble methods, estimate uncertainty by sampling probability predictions from different submodels, which leads to slow uncertainty estimation at inference time. Recent works address this drawback by directly predicting parameters of prior distributions over the probability predictions with a neural network. While this approach has demonstrated accurate uncertainty estimation, it requires defining arbitrary target parameters for in-distribution data and makes the unrealistic assumption that out-of-distribution (OOD) data is known at training time.

In this work we propose the Posterior Network (PostNet), which uses Normalizing Flows to predict an individual closed-form posterior distribution over predicted probabilites for any input sample. The posterior distributions learned by PostNet accurately reflect uncertainty for in- and out-of-distribution data – without requiring access to OOD data at training time. PostNet achieves state-of-the art results in OOD detection and in uncertainty calibration under dataset shifts.

## 1   Introduction

Quantifying uncertainty in neural network predictions is key to making Machine Learning reliable. In many sensitive domains, like robotics, financial or medical areas, giving autonomy to AI systems is highly dependent on the trust we can assign to them. In addition, AI systems being aware about their predictions' uncertainty, can adapt to new situations and refrain from taking decisions in unknown or unsafe conditions. Despite of this necessity, traditional neural networks show overconfident predictions, even for data that is signifanctly different from the training data [21] [11]. In particular, they should distinguish between *aleatoric* and *epistemic* uncertainty, also called data and knowledge uncertainty [24], respectively. The aleatoric uncertainty is irreducible from the data, e.g. a fair coin has 50/50 chance for head. The epistemic uncertainty is due to the lack of knowledge about unseen data, e.g. an image of an unknown object or an outlier in the data.

**Related work.** Uncertainty estimation is a growing research area unifying various approaches [3, 23, 30, 7, 21, 8]. Bayesian Neural Networks learn a *distribution* over the weights [3, 23, 30]. Another class of approaches uses a collection of sub-models and aggregates their predictions, which are in turn used to estimate statistics (e.g., mean and variance) of the class probability distribution. Even though such methods, like ensemble and drop-out, have demonstrated remarkable performance (see, e.g., [38]), they describe implicit distributions for predictions and require a costly sampling phase at inference time for uncertainty estimation.

Recently, a new class of models aims to directly predict the parameters of a prior distribution on the categorical probability predictions, accounting for the different types of uncertainty [24, 25, 35, 1]. However, these methods require (i) the definition of arbitrary target prior distributions [24, 25, 35],

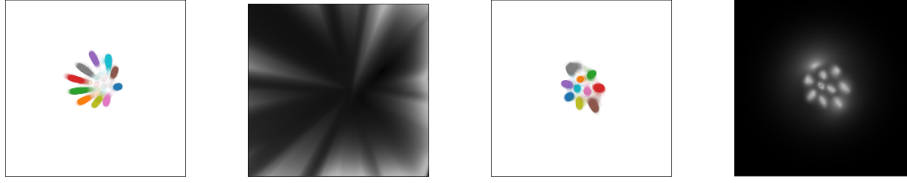

(a) Data labels - PriorNet(b) Uncertainty - PriorNet (c) Data labels - PostNet (d) Uncertainty - PostNet

Figure 1: PriorNet has a pre-ultimate layer of dimension 2 and was trained with Reverse KL and uniform noise on $[0, 255]^{28 \times 28}$ as OOD data. PostNet has a latent space of dimension 2 and was trained without OOD data. (a) and (c) show the learned latent positions of data with colored labels. (b) and (d) show uncertainty estimates in the latent spaces where darker regions indicate high uncertainty. PostNet correctly assigns high uncertainty to OOD regions contrary to PriorNet.

and most importantly, (ii) out-of-distribution (OOD) samples during training time, which is an unrealistic assumption in most applications [24, 25].

Classically, these models would use both ID and OOD samples (e.g. MNIST and FashionMNIST) during training to detect similar OOD samples (i.e. FashionMNIST) at inference time. We show that preventing access to explicit OOD data during training leads to poor results using these approaches (see Fig. 1 for MNIST; or app. for toy datasets). Contrary to the expected results, these models produce increasingly confident predictions for samples far from observed data. In contrast, we propose Posterior Network (PostNet), which assigns high epistemic uncertainty to out-of-distribution samples, low overall uncertainty to regions nearby observed data of a single class, and high aleatoric and low epistemic uncertainty to regions nearby observed data of different classes.

PostNet uses normalizing flows to learn a distribution over Dirichlet parameters in latent space. We enforce the densities of the individual classes to integrate to the number of training samples in that class, which matches well with the intuition of Dirichlet parameters corresponding to the number of observations per class. PostNet does not require any OOD samples for training, the (arbitrary) specification of target prior distributions, or costly sampling for uncertainty estimation at test time.

## 2   Posterior Network

In classification, we can distinguish between two types of uncertainty for a given input $\mathbf{x}^{(i)}$: the uncertainty on the class prediction $y^{(i)} \in \{1, \ldots, C\}$ (i.e. aleatoric unceratinty), and the uncertainty on the categorical distribution prediction $\mathbf{p}^{(i)} = [p_1^{(i)}, \ldots, p_C^{(i)}]$ (i.e. epistemic uncertainty). A convenient way to model both is to describe the *epistemic distribution* $q^{(i)}$ of the categorical distribution prediction $\mathbf{p}^{(i)}$, i.e. $\mathbf{p}^{(i)} \sim q^{(i)}$. From the epistemic distribution follows naturally an estimate of the *aleatoric distribution* of the class prediction $y^{(i)} \sim \text{Cat}(\bar{\mathbf{p}}^{(i)})$ where $\mathbb{E}_{q^{(i)}}[\mathbf{p}^{(i)}] = \bar{\mathbf{p}}^{(i)}$.

Approaches like ensembles [21] and dropout [8] model $q^{(i)}$ implicitly, which only allows them to estimate statistics at the cost of $S$ samples (e.g. $\mathbb{E}_{q^{(i)}}[\mathbf{p}^{(i)}] \approx \frac{1}{S} \sum_{s=1}^{S} \tilde{\mathbf{p}}^{(i)}$ where $\tilde{\mathbf{p}}^{(i)}$ is sampled from $q^{(i)}$). Another class of models [24, 25, 1, 35] explicitly parametrizes the epistemic distribution with a Dirichlet distribution (i.e. $q^{(i)} = \text{Dir}(\boldsymbol{\alpha}^{(i)})$ where $f_\theta(x^{(i)}) = \boldsymbol{\alpha}^{(i)} \in \mathbb{R}_+^C$), which is the natural prior for categorical distributions. This parametrization is convenient since it requires only one pass to compute epistemic distribution, aleatoric distribution and class prediction:

$$q^{(i)} = \text{Dir}(\boldsymbol{\alpha}^{(i)}), \qquad \bar{p}_c^{(i)} = \frac{\alpha_c}{\alpha_0} \text{ with } \alpha_0 = \sum_{c=1}^{C} \alpha_c, \qquad y^{(i)} = \arg\max \left[\bar{p}_1, ..., \bar{p}_C\right] \qquad (1)$$

The concentration parameters $\alpha_c^{(i)}$ can be interpreted as the number of observed samples of class $c$ and, thus, are a good indicator of epistemic uncertainty for non-degenerate Dirichlet distributions (i.e. $\alpha_c^{(i)} \geq 1$). To learn these parameters, Prior Networks [24, 25] use OOD samples for training and define different target values for ID and OOD data. For ID data, $\alpha_c^{(i)}$ is set to an arbitrary, large number if $c$ is the correct class and 1 otherwise. For OOD data, $\alpha_c^{(i)}$ is set to 1 for all classes.

This approach has four issues: **(1)** The knowledge of OOD data for training is unrealistic. In practice, we might not have these data, since OOD samples are by definition not likely to be observed. **(2)** Discriminating in- from out-of-distribution data by providing an explicit set of OOD samples

is hopeless. Since any data not from the data distribution is OOD, it is therefore impossible to characterize the infinitely large OOD distribution with an explicit data set. **(3)** The predicted Dirichlet parameters can take any value, especially for new OOD samples which were not seen during training. In the same way, the sum of the total fictitious prior observations over the full input domain $\int \alpha_0(\mathbf{x})d\mathbf{x}$ is not bounded and in particular can be much larger than the number of ground-truth observations $N$. This can result in undesired behavior and assign arbitrarily high epistemic certainty for OOD data not seen during training. **(4)** Besides producing such arbitrarily high prior confidence, PNs can also produce degenerate concentration parameters (i.e. $\alpha_c < 1$). While [25] tried to fix this issue by using a different loss, nothing intrinsically prevents Prior Networks from predicting degenerate prior distributions. In the following section we describe how Posterior Network solves these drawbacks.

## 2.1  An input-dependent Bayesian posterior

First, recall the Bayesian update of a single categorical distribution $y \sim \text{Cat}(\mathbf{p})$. It consists in (1) introducing a prior Dirichlet distribution over its parameters i.e. $\mathbb{P}(\mathbf{p}) = \text{Dir}(\boldsymbol{\beta}^{\text{prior}})$ where $\boldsymbol{\beta}^{\text{prior}} \in \mathbb{R}_+^C$, and (2) using $N$ given observations $y^{(1)}, ..., y^{(N)}$ to form the posterior distribution $\mathbb{P}(\mathbf{p}|\{y^{(j)}\}_{j=1}^N) = \text{Dir}(\boldsymbol{\beta}^{\text{prior}} + \boldsymbol{\beta}^{\text{data}})$ where $\beta_c^{\text{data}} = \sum_j \mathbb{1}_{y^{(j)}=c}$ are the class counts. That is, the Bayesian update is

$$\mathbb{P}(\mathbf{p}|\{y^{(j)}\}_{j=1}^n) \propto \mathbb{P}(\{y^{(j)}\}_{j=1}^n|\mathbf{p}) \times \mathbb{P}(\mathbf{p}). \tag{2}$$

Observing no data (i.e. $\beta_c^{\text{data}} \rightarrow 0$) would lead to flat categorical distribution (i.e. $p_c = \beta_c^{\text{prior}} \cdot (\sum_{c'} \beta_{c'}^{\text{prior}})^{-1}$), while observing many samples (i.e. $\beta_c^{\text{data}}$ is large) would converge to the true data distribution (i.e. $p_c \approx \frac{\beta_c}{\sum_i \beta_i}$). Furthermore, we remark that $N$ behaves like a certainty budget distributed over all classes i.e. $N = \sum_c \beta_c^{\text{data}}$.

Classification is more complex. Generally, we predict the class label $y^{(i)}$ from a different categorical distribution $\text{Cat}(\mathbf{p}^{(i)})$ for each input $\mathbf{x}^{(i)}$. PostNet extends the Bayesian treatment of a single categorical distribution to classification by predicting an individual posterior update for any possible input. To this end, it distinguishes between a fixed prior parameter $\boldsymbol{\beta}^{\text{prior}}$ and the additional learned (pseudo) counts $\boldsymbol{\beta}^{(i)}$ to form the parameters of the posterior Dirichlet distribution $\boldsymbol{\alpha}^{(i)} = \boldsymbol{\beta}^{\text{prior}} + \boldsymbol{\beta}^{(i)}$. Hence, PostNet's posterior update is equivalent to predicting a set of pseudo observations $\{\tilde{y}^{(j)}\}_j^{(i)}$ per input $\mathbf{x}^{(i)}$, accordingly $\beta_c^{(i)} = \sum_j \mathbb{1}_{\tilde{y}^{(j)}=c}$ and

$$\mathbb{P}(\mathbf{p}^{(i)}|\{\tilde{y}^{(j)}\}_j^{(i)}) \propto \mathbb{P}(\{\tilde{y}^{(j)}\}_j^{(i)}|\mathbf{p}^{(i)}) \times \mathbb{P}(\mathbf{p}^{(i)}). \tag{3}$$

In practice, we set $\boldsymbol{\beta}^{\text{prior}} = \mathbf{1}$ leading to a flat equiprobable prior when the model brings no additional evidence, i.e. when $\boldsymbol{\beta}^{(i)} = \mathbf{0}$.

The parametrization of $\beta_c^{(i)}$ is crucial and based on two main components. The first component is an encoder neural network, $f_\theta$ that maps a data point $\mathbf{x}^{(i)}$ onto a low-dimensional latent vector $\mathbf{z}^{(i)} = f_\theta(\mathbf{x}^{(i)}) \in \mathbb{R}^H$. The second component is to learn a *normalized* probability density $\mathbb{P}(\mathbf{z}|c; \phi)$ per class on this latent space; intuitively acting as class conditionals in the latent space. Given these and the number of ground-truth observations $N_c$ in class $c$, we define:

$$\beta_c^{(i)} = N_c \cdot \mathbb{P}(\mathbf{z}^{(i)}|c; \phi) = N \cdot \mathbb{P}(\mathbf{z}^{(i)}|c; \phi) \cdot \mathbb{P}(c), \tag{4}$$

which corresponds to the number of (pseudo) observations of class $c$ at $\mathbf{z}^{(i)}$. Note that it is crucial that $\mathbb{P}(\mathbf{z}|c; \phi)$ corresponds to a proper normalized density function since this will ensure the model's epistemic uncertainty to increase outside the known distribution. Indeed, the core idea of our approach is to parameterize these distributions by a flexible, yet tractable family: normalizing flows (e.g. radial flow [34] or IAF [17]). Note that normalizing flows are theoretically capable of modeling any continuous distribution given an expressive and deep enough model [14, 17].

In practice, we observed that various architectures can be used for the encoder. Also, similarly to GAN training [33], we observed that adding a batch normalization after the encoder made the training more stable. It facilitates the match between the latent positions output by the encoder and non-zero density regions learned by the normalizing flows. Remark that we can theoretically use any density estimator on the latent space. We experimentally compared Mixtures of Gaussians (MoG), radial flow [34] and IAF [17]. While all density types performed reasonably well (see Tab. 1 and app.), we observed a better performance of flow-based density estimation in general. We decided to use radial flow for its good trade-off between flexibility, stability, and compactness (only few parameters).

**Model discussion.** Equation 4 exhibits a set of interesting properties which ensure reasonable uncertainty estimates for ID and OOD samples. To highlight the properties of the Dirichlet distributions learned by PostNet, we assume in this paragraph that we have a fixed encoder $f_\theta$ and normalizing flow model parameterized by $\phi$. Writing the mean of the Dirichlet distribution parametrized by (4) and using Bayes' theorem gives:

$$\mathbb{E}_{\mathbf{p}\sim\text{Dir}(\boldsymbol{\alpha}^{(i)})}[p_c] = \frac{\beta_c^{\text{prior}} + N \cdot \mathbb{P}(c|\mathbf{z}^{(i)};\phi) \cdot \mathbb{P}(\mathbf{z}^{(i)};\phi)}{\sum_c \beta_c^{\text{prior}} + N \cdot \mathbb{P}(\mathbf{z}^{(i)};\phi)} \tag{5}$$

For very likely in-distribution data (i.e. $\mathbb{P}(\mathbf{z}^{(i)};\phi) \to \infty$), the aleatoric distribution estimate $\bar{\mathbf{p}}^{(i)} = \mathbb{E}_{\mathbf{p}\sim\text{Dir}(\boldsymbol{\alpha}^{(i)})}[p_c]$ converges to the true categorical distribution $\mathbb{P}(c|\mathbf{z}^{(i)};\phi)$. Thus, predictions are more accurate and calibrated for likely samples. Conversely, for out-of-distribution samples (i.e. $\mathbb{P}(\mathbf{z}^{(i)};\phi) \to 0$), the aleatoric distribution estimate $\bar{\mathbf{p}}^{(i)} = \mathbb{E}_{\mathbf{p}\sim\text{Dir}(\boldsymbol{\alpha}^{(i)})}[p_c]$ converges to the flat prior distribution (e.g. $p_c = \frac{1}{C}$ if $\boldsymbol{\beta}^{prior} = \mathbf{1}$). In the same way, we show in the appendix that the covariance of the epistemic distribution converges to $\mathbf{0}$ for very likely in-distribution data, meaning no epistemic uncertainty. Thus, uncertainty for in-distribution data is reduced to the (inherent) aleatoric uncertainty and zero epistemic uncertainty. Similarly to the single categorical distribution case, increasing the training dataset size (i.e. $N \to \infty$) also leads to the mean prediction $\mathbb{E}_{\mathbf{p}\sim\text{Dir}(\boldsymbol{\alpha})}[p_c]$ converging to the class posterior $\mathbb{P}(c|\mathbf{z}^{(i)};\phi)$. On the contrary, no observed data (i.e $N = 0$) again leads to the model reverting to a flat prior.

Posterior Network also handles limited certainty budgets at different levels. At the sample level, the certainty budget $\alpha_0^{(i)} = \sum_c \alpha_c^{(i)}$ is distributed over classes. At the class level, the certainty budget $N_c = \int N_c\, \mathbb{P}(\mathbf{z}|c;\phi)\, d\mathbf{z} = N_c \int \mathbb{P}(\mathbf{z}|c;\phi)\, d\mathbf{z}$ is distributed over samples. At the dataset level, the certainty budget $N = \sum_c \int N_c\, \mathbb{P}(\mathbf{z}|c;\phi)\, d\mathbf{z}$ is distributed over classes and samples. Regions of latent space with many training examples are assigned high density $\mathbb{P}(\mathbf{z}|c;\phi)$, forcing low density elsewhere to fulfill the integration constraint. Consequently, density estimation using normalizing flows enables PostNet to learn out-of-distribution uncertainty by observing only in-distribution data.

**Overview.** In Figure 2 we provide an overview of Posterior Network. We have three example inputs, $\mathbf{x}^{(1)}$, $\mathbf{x}^{(2)}$, and $\mathbf{x}^{(3)}$, which are mapped onto their respective latent space coordinates $\mathbf{z}^{(i)}$ by the encoding neural network $f_\theta$. The normalizing flow component learns flexible (normalized) density functions $\mathbb{P}(\mathbf{z}|c;\phi)$, for which we evaluate their densities at the positions of the latent vectors $\mathbf{z}^{(i)}$. These densities are used to parameterize a Dirichlet distribution for each data point, as seen on the right hand side. Higher densities correspond to higher confidence in the Dirichlet distributions – we can observe that the out-of-distribution sample $\mathbf{x}^{(3)}$ is mapped to a point with (almost) no density, and hence its predicted Dirichlet distribution has very high epistemic uncertainty. On the other hand, $\mathbf{x}^{(2)}$ is an ambiguous example that could depict either the digit 0 or 6. This is reflected in its corresponding Dirichlet distribution, which has high aleatoric uncertainty (as the sample is ambiguous), but low epistemic uncertainty (since it is from the distribution of hand-drawn digits). The unambiguous sample $\mathbf{x}^{(1)}$ has low overall uncertainty.

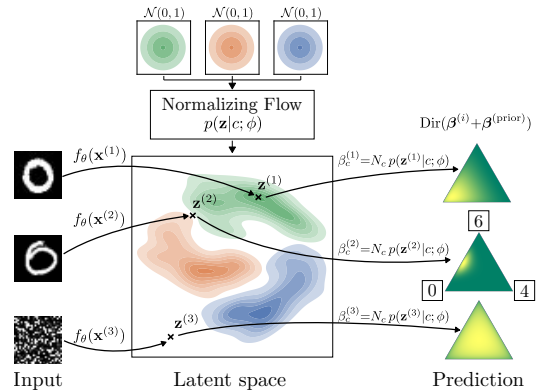

Figure 2: Overview of Posterior Network.

Lastly, since both the encoder network $f_\theta$ and the normalizing flow parameterized by $\phi$ are fully differentiable, we can learn their parameters jointly in an end-to-end fashion. We do this via a novel loss defined in Sec.3 which emerges from Bayesian learning principles [36] and is related to UCE [1].

## 2.2 Density estimation for OOD detection

Normalized densities, as used by PostNet, are well suited to discriminate between ID data (with high likelihoood) and OOD data (with low likelihood). While it is also possible to learn a normalizing flow model on the input domain directly (e.g., [31, 15, 9]), this is very computationally demanding and might not be necessary for a discriminative model. Futhermore, density estimation is prone to the curse of dimensionality in high dimensional spaces [27, 4]. For example, unsupervised deep

generative models like [16] or [39] have been shown to be unable to distinguish between ID and OOD samples in some situations when working on all features directly [28, 10].

To circumvent these issues, PostNet leverages two techniques. First, it uses the full class label information. Thus, PostNet assigns a density per class and regularizes the epistemic uncertainty with the training class counts $N_c$. Second, PostNet performs density estimation on a low dimensional latent space describing relevant features for the classification task (Fig. 1; Fig. 2). Hence, PostNet does not suffer from the curse of dimensionality and still enjoys the benefits of a properly normalized density function. Using the inductive bias of a discriminative task and low dimensional latent representations improved OOD detection in [18] as well.

## 3   Uncertainty-Aware Loss Computation

A crucial design choice for neural network learning is the loss function. PostNet estimates both aleatoric and epistemic uncertainty by learning a distribution $q^{(i)}$ for data point $i$ which is close to the true posterior of the categorical distribution $\mathbf{p}^{(i)}$ given the training data $\mathbf{x}^{(i)}$: $q(\mathbf{p}^{(i)}) \simeq \mathbb{P}(\mathbf{p}^{(i)}|\mathbf{x}^{(i)})$. One way to approximate the posterior is the following Bayesian loss function [2, 36, 41], which has the nice property of being optimal when $q^{(i)}$ is equal to the true posterior distribution:

$$q^* = \underset{q^{(i)} \in \mathcal{P}}{\arg\min} \; \mathbb{E}_{\psi^{(i)} \sim q(\psi^{(i)})}[l(\psi^{(i)}, \mathbf{x}^{(i)})] - H(q^{(i)}), \tag{6}$$

where $l$ is a generic loss over $\psi^{(i)}$ satisfying $0 < \int \exp(-l(\psi, x))d\psi < \infty$, $\mathcal{P}$ is the family of distributions we consider and $H(q^{(i)})$ denotes the entropy of $q^{(i)}$.

Applied in our case, Posterior Network learns a distribution $q$ from the family of the Dirichlet distributions $\mathrm{Dir}(\boldsymbol{\alpha}^{(i)}) = \mathcal{P}$ over the parameters $\mathbf{p}^{(i)} = \psi^{(i)}$. Instantiating the loss $l$ with the cross-entropy loss (CE) we obtain the following optimization objective

$$\min_{\theta,\phi} \mathcal{L} = \min_{\theta,\phi} \frac{1}{N} \sum_i^N \underbrace{\mathbb{E}_{q(\mathbf{p}^{(i)})}[\mathrm{CE}(\mathbf{p}^{(i)}, \mathbf{y}^{(i)})]}_{(1)} - \underbrace{H(q^{(i)})}_{(2)} \tag{7}$$

where $\mathbf{y}^{(i)}$ corresponds to the one-hot encoded ground-truth class of data point $i$. Optimizing this loss approximates the true posterior distribution for the the categorical distribution $\mathbf{p}^{(i)}$. The first term (1) corresponds the Uncertain Cross Entropy loss (UCE) introduced by [1], which is known to increase confidence for observed data. The second term (2) is an entropy regularizer, which emerges naturally and favors smooth distributions $q^{(i)}$. Here we are optimizing jointly over the neural network parameters $\theta$ and $\phi$. We also experimented with sequential training i.e. optimizing over the normalizing flow component only with a pre-trained model (see Tab. 2 and app.). Once trained, PostNet can predict uncertainty-aware Dirichlet distributions for unseen data points.

Observe that Eq. (7) is equivalent to the ELBO loss used in variational inference when using a uniform Dirichlet prior (i.e. $(1) = -\mathbb{E}_{q(\mathbf{p}^{(i)})}[\log \mathbb{P}(\mathbf{y}^{(i)}|\mathbf{p}^{(i)})]$ and $(2) = \mathrm{KL}(q^{(i)}||\mathbb{P}(\mathbf{p}^{(i)}))$ where $\mathbb{P}(\mathbf{p}^{(i)}) = \mathrm{Dir}(\mathbf{1})$). The more general Eq. (6), however, is not necessarily equal to an ELBO loss.

Another interesting special case is to consider the family of Dirac distributions as $\mathcal{P}$ instead of the family of Dirichlet distributions. In this case we find back the traditional cross-entropy loss, which performs a simple point estimate for the distribution $\mathbf{p}^{(i)}$. CE is therefore not suited to learn a distribution with non-zero variance, as explained in [1].

Other approaches, such as dropout, approximate the expectation in UCE by sampling from $q^{(i)}$. Our approach has the advantage of using closed-form expressions both for UCE [1] and the entropy term, thus being efficient and exact. The weight of the entropy regularization is a hyperparameter; experiments have shown PostNet to be fairly insensitive to it, so in our experiments we set it to $10^{-5}$.

## 4   Experimental Evaluation

In this section we compare our model to previous methods on a rich set of experiments. The code and further supplementary material is available online (www.daml.in.tum.de/postnet).

**Baselines.** We have special focus on comparing with other models parametrizing Dirichlet distributions. We use Prior Networks (PN) trained with KL divergence (**KL-PN**) [24] and Reverse KL divergence (**RKL-PN**) [25]. These methods assume the knowledge of in- and out-of-distribution samples. For fair evaluation, the actual OOD test data cannot be used; instead, we used uniform noise on the valid domain as OOD training data. Additionally we trained RKL-PN with FashionMNIST as OOD data for MNIST (**RKL-PN w/ F.**). We also compare to Distribution Distillation (**Distill.**) [26], which learns Dirichlet distributions with maximum likelihood by using soft-labels from an ensemble of networks. As further baselines, we compare to dropout models (**Dropout Net**) [8] and ensemble methods (**Ensemble Net**) [21], which are state of the art in many tasks involving uncertainty estimation [38]. Empirical estimates of the mean and variance of $q^{(i)}$ are computed based on the neuron drop probability $p_{\text{drop}}$, and $m$ individually trained networks for ensemble.

All models share the same core architecture using 3 dense layers for tabular data, and 3 conv. + 3 dense layers for image data. Similarly to [24, 25], we also used the VGG16 architecture [37] on CIFAR10. We performed a grid search on $p_{\text{drop}}$, $m$, learning rate and hidden dimensions, and report results for the best configurations. Results are obtained from 5 trained models with different initializations. Moreoever, for all experiments, we split the data into train, validation and test set ($60\%, 20\%, 20\%$) and train/evaluate all models on 5 different splits. Besides the mean we also report the standard error of the mean. Further details are given in the appendix.

**Datasets.** We evaluate on the following real-world datasets: **Segment** [6], **Sensorless Drive** [6], **MNIST** [22] and **CIFAR10** [19]. The former two datasets (Segment and Sensorless Drive) are tabular datasets with dimensionality 18 and 49 and with 7 and 11 classes, respectively. We rescale all inputs between $[0, 1]$ by using the min and max value of each dimension from the training set. Additionally, we compare all models on 2D synthetic data composed of three Gaussians each. Datasets are presented in more detail in the appendix.

**Uncertainty Metrics.** We follow the method proposed in [38] and evaluate the coherence of confidence, uncertainty calibration and OOD detection. Note that our goal is not to improve accuracy; still we report the numbers in the experiments.

Confidence calibration: We aim to answer '*Are more confident (i.e. less uncertain) predictions more likely to be correct?*'. We use the area under the precision-recall curve (AUC-PR) to measure confidence calibration. For aleatoric confidence calibration (**Alea. Conf.**) we use $\max_c \bar{\mathbf{p}}_c^{(i)}$ as the scores with labels 1 for correct and 0 for incorrect predictions. For epistemic confidence calibration (**Epist. Conf.**), we distinguish Dirichlet-based models, and dropout and ensemble models. For the former we use $\max_c \boldsymbol{\alpha}_c^{(i)}$ as scores, and for the latter we use the (inverse) empirical variance $\tilde{\mathbf{p}}_c^{(i)}$ of the predicted class, estimated from 10 samples.

Uncertainty calibration: We used Brier score (**Brier**), which is computed as $\frac{1}{N} \sum_i^N \|\bar{\mathbf{p}}^{(i)} - \mathbf{y}^{(i)}\|_2$, where $\mathbf{y}^{(i)}$ is the one-hot encoded ground-truth class of data point $i$. For Brier score, lower is better.

OOD detection: Our main focus lies on the models' ability to detect out-of-distribution samples. We used AUC-PR to measure performance. For aleatoric OOD detection (**Alea. OOD**), the scores are $\max_c \bar{\mathbf{p}}_c^{(i)}$ with labels 1 for ID data and 0 for OOD data. Fo epistemic OOD detection (**Epist. OOD**), the scores for Dirichlet-based models are given by $\alpha_0^{(i)} = \sum_c \alpha_c^{(i)}$, while we use the (inverse) empirical variance $\tilde{\mathbf{p}}^{(i)}$ for ensemble and dropout models. To provide a comprehensive overview of OOD detection results we use different types of OOD data as described below.

*Unseen datasets.* We use data from other datasets as OOD data for the image-based models. We use data from FashionMNIST [40] and K-MNIST [5] as OOD data for models trained on MNIST, and data from SVHN [29] as OOD for CIFAR10.

*Left-out classes.* For the tabular datasets (Segment and Sensorless Drive) there are no other datasets that are from the same domain. To simulate OOD data we remove one or more classes from the training data and instead consider them as OOD data. We removed one class (class sky) from the Segment dataset and two classes from Sensorless Drive (class 10 and 11).

*Out-of-domain.* In this novel evaluation we consider an extreme case of OOD data for which the data comes from different value ranges (**OODom**). E.g., for images we feed unscaled versions in the range $[0, 255]$ instead of scaled versions in $[0, 1]$. We argue that models should easily be able to detect data that is extremely far from the data distribution. However, as it turns out, this is surprisingly difficult for many baseline models.

|  | Acc. | Alea. Conf. | Epist. Conf. | Brier | OOD Alea. | OOD Epist. |
|---|---|---|---|---|---|---|
| **Drop Out** | 89.32±0.2 | 98.21±0.1 | 95.24±0.2 | 28.86±0.4 | 35.41±0.4 | 40.61±0.7 |
| **Ensemble** | 99.37±0.0 | 99.99±0.0 | *99.98±0.0 | 2.47±0.1 | 50.01±0.0 | 50.62±0.1 |
| **Distill.** | 93.66±1.5 | 98.29±0.5 | 98.15±0.5 | 44.94±1.4 | 32.1±0.6 | 31.17±0.2 |
| **KL-PN** | 94.77±0.9 | 99.52±0.1 | 99.47±0.1 | 21.47±1.9 | 35.48±0.8 | 33.2±0.6 |
| **RKL-PN** | 99.42±0.0 | 99.96±0.0 | 99.89±0.0 | 9.07±0.1 | 45.89±1.6 | 38.14±0.8 |
| **PostN Rad.** | 98.02±0.1 | 99.89±0.0 | 99.47±0.0 | 5.51±0.2 | 72.89±0.8 | *88.73±0.5 |
| **PostN IAF** | *99.52±0.0 | *100.0±0.0 | 99.92±0.0 | *1.43±0.1 | *82.96±0.8 | 88.65±0.4 |

Table 1: Results on Sensorless Drive dataset. Bold numbers indicate best score among Dirichlet parametrized models and starred numbers indicate best scores among all models.

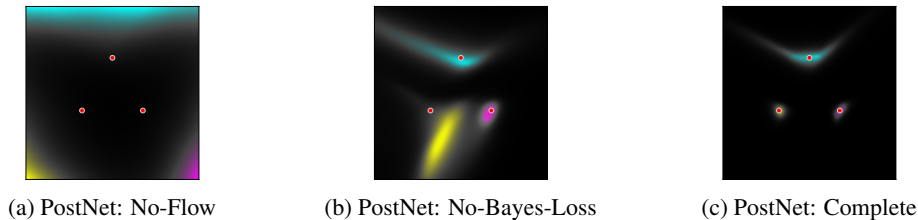

(a) PostNet: No-Flow        (b) PostNet: No-Bayes-Loss        (c) PostNet: Complete

Figure 3: Uncertainty visualization for a 2D 3-Gaussians dataset. Red dots indicate the Gaussians means. Darker regions indicate high epistemic uncertainty for a class prediction. Ablated models fail even a simple dataset while PostNet shows high certainty around gaussians means only.

*Dataset shifts.* Finally, for CIFAR10, we use 15 different image corruptions at 5 different severity levels [13]. This setting evaluates the models' ability to detect low-quality data (Fig. 4b,c).

**Results.** Results for the Sensorles Drive dataset are shown in Tab. 1. Tables for other datasets are in the appendix. Even without requiring expensive sampling, PostNet performs on par for accuracy and confidence scores with other models, brings a significant improvement for calibration within the Dirichlet-based models, and outperforms all other models by a large margin (more than $+30\%$ abs. improvement) for OOD detection. Radial flow and IAF variants both achieve strong performance for all datasets (see app.). We use the smaller model (i.e. Radial flow) for comparison in the following. In our experiments, note that using one Radial flow per class represents a small overhead of only $80$ parameters per class, which is negligible compared to the encoder architectures (e.g. VGG16 has 138M parameters).

|  | Acc. | Alea. Conf. | Epist. Conf. | Brier | OOD Alea. | OOD Epist. |
|---|---|---|---|---|---|---|
| **PostN: No-Flow** | 55.38±0.7 | 85.46±0.3 | 82.58±0.6 | 64.4±0.6 | 29.59±0.1 | 31.15±0.4 |
| **PostN: No-Bayes-Loss** | 96.6±0.2 | 99.74±0.0 | 98.68±0.1 | 8.85±0.4 | 62.39±1.5 | 82.63±1.4 |
| **PostN: Seq-No-Bn** | 15.09±1.0 | 39.88±7.2 | 39.86±7.2 | 89.88±1.3 | 57.19±2.5 | 56.74±2.4 |
| **PostN: Seq-Bn** | 98.42±0.1 | 99.92±0.0 | 98.76±0.1 | 5.41±0.1 | 52.35±0.7 | 71.75±1.9 |

Table 2: Ablation study results on Sensorless Drive dataset. Gray cells indicate significant drops in scores compared to complete PostNet Rad. in Tab. 1.

We performed an ablation study on each component of PostNet to evaluate their individual contributions. We were especially interested in comparing stability and uncertainty estimates. Thus, we removed independently the normalizing flow component (No-Flow) and the novel Bayesian loss (No-Bayes-Loss) replaced by the classic cross-entropy loss. Furthermore, we used pre-trained models and subsequently only trained the normalizing flow component, with or without a batch normalization layer (Seq-Bn and Seq-No-Bn). We report results in Tab. 2. No-Flow has a significant drop in OOD detection scores similarly to Prior Networks; not surprising since they mainly differ by their loss. This underlines the importance of using normalized density estimation to differentiate ID and OOD data. The lower performance of No-Bayes-Loss compared to the original model indicates the benefit of using our Bayesian loss. Seq-Bn obtains good performance for some of the metrics, which as a by-product, allows to estimate uncertainty on pre-trained models. Though, we noticed better performance for joint training in general. As shown by Seq-No-Bn scores, the batch normalization

layer brings stability. It intuitively facilitates predicted latent positions to lie on non-zero density regions. Similar conclusions can be drawn on the toy dataset (see Fig. 3) and the Segment dataset (see app.). We further compare various density types and latent dimensions in appendix. We noticed that a too high latent dimension leads to a performance decrease. We also observed that flow-based density estimation generally achieves better scores.

| | OOD K. Alea. | OOD K. Epist. | OOD F. Alea. | OOD F. Epist. | OODom K. Alea. | OODom K. Epist. | OODom F. Alea. | OODom F. Epist. |
|---|---|---|---|---|---|---|---|---|
| **RKL-PN** | 60.76±2.9 | 53.76±3.4 | 78.45±3.1 | 72.18±3.6 | 9.35±0.1 | 8.94±0.0 | 9.53±0.1 | 8.96±0.0 |
| **RKL-PN w/ F.** | 81.34±4.5 | 78.07±4.8 | **100.0±0.0** | **100.0±0.0** | 9.24±0.1 | 9.08±0.1 | 88.96±4.4 | 87.49±5.0 |
| **PostN** | **95.75±0.2** | **94.59±0.3** | 97.78±0.2 | 97.24±0.3 | **100.0±0.0** | **100.0±0.0** | **100.0±0.0** | **100.0±0.0** |

Table 3: Results on MNIST for OOD detection against KMNIST (K.) and FashionMNIST (F.). We trained Rev. KL divergence PriorNets with uniform noise (RKL-PN) and Fashion MNIST (RKL-PN w/ F.) as OOD. PostNet requires no OOD data. Larger numbers are better.

Results of the comparison between RKL-PN, RKL-PN w/ F and PostNet for OOD detection on MNIST are shown in Tab. 3. Not surprisingly, the usage of FashionMNIST as OOD data for training helped RKL-PN to detect other FashionMNIST data. Except for FashionMNIST OOD, PostNet still outperforms RKL-PN w/ F. in OOD detection for other datasets. We noticed that tabular datasets, defined on an unbounded input domain, are more difficult for baselines. One explanation is that due to the $\min/\max$ normalization it can happen that test samples lie outside the interval $[0, 1]$ observed during training. For images, the input domain is compact, which allows to define a valid distribution for OOD data (e.g. uniform) which makes OODom data challenging (see OOD vs OODom in Tab. 3).

| | Acc. | Alea. Conf. | Epist. Conf. | Brier | OOD Alea. | OOD Epist. | OODom Alea. | OODom Epist. |
|---|---|---|---|---|---|---|---|---|
| **Drop Out C.** | 71.73±0.2 | 92.18±0.1 | 84.38±0.3 | 49.76±0.2 | **72.94±0.3** | 41.68±0.5 | 28.3±1.8 | 47.1±3.3 |
| **KL-PN C.** | 48.84±0.5 | 78.01±0.6 | 77.99±0.7 | 83.11±0.6 | 59.32±1.1 | 58.03±0.8 | 17.79±0.0 | 20.25±0.2 |
| **RKL-PN C.** | 62.91±0.3 | 85.62±0.2 | 81.73±0.2 | 58.12±0.4 | 67.07±0.4 | 56.64±0.8 | 17.76±0.0 | 17.76±0.0 |
| **PostN C.** | **76.46±0.3** | **94.75±0.1** | **94.34±0.1** | **37.39±0.4** | 72.83±0.6 | **72.82±0.7** | **100.0±0.0** | **100.0±0.0** |
| **Drop Out V.** | 82.84±0.1 | 97.15±0.0 | 96.6±0.0 | 27.15±0.2 | 51.39±0.1 | 53.64±0.1 | 51.38±0.1 | 53.66±0.1 |
| **KL-PN V.** | 27.46±1.7 | 50.61±4.0 | 52.49±4.2 | 87.28±1.0 | 43.96±1.9 | 43.23±2.3 | 18.14±0.1 | 19.12±0.4 |
| **RKL-PN V.** | 64.76±0.3 | 86.11±0.4 | 85.59±0.3 | 54.73±0.4 | 53.61±1.1 | 49.37±0.8 | 29.07±2.1 | 24.84±1.3 |
| **PostN V.** | **84.85±0.0** | **97.76±0.0** | **97.25±0.0** | **22.84±0.0** | **80.21±0.2** | **77.71±0.3** | **91.35±0.5** | **99.25±0.1** |

Table 4: Results on CIFAR10 with simple convolutional architectures (C.) and VGG16 (V.). Bold numbers indicate best score among one architecture type.

Uncertainty estimation should be good regardless of the model accuracy. It is even more important for less accurate models since they actually *do not know* (i.e. they do more mistakes). Thus, we compared the models that use a single network for training (using a convolutional architecture and VGG16) in Tab. 4. Without the knowledge of true OOD data (SVHN) during training, Prior Networks struggle to achieve good performance. In contrast, PostNet outputs high quality uncertainty estimates regardless of the architecture used for the encoder. We report additional results for PostNet using other encoder architectures (convolutional architecture, AlexNet [20], VGG [37] and ResNet [12]) in Table 5. Deep generative models as Glow [16] using density estimation on input space are unable to distinguish between CIFAR10 and SVHN [28]. In contrast, PostNet clearly distinguishes between in-distribution data (CIFAR10) with low entropy, out-of-distribution (OOD SVHN) with high entropy, and close to the maximum possible entropy for out-of-domain data (OODom SVHN) (see Fig. 4a). Similar conclusions hold for MNIST and FashionMNIST (see app.). Furthermore, results for the image perturbations on CIFAR10 introduced by [13] are presented in Fig. 4. We define the average change in confidence as the ratio between the average confidence $\frac{1}{N} \sum_i^N \alpha_0^{(i)}$ at severity 1 vs other severity levels. As larger shifts correspond to larger differences in the underlying distributions, we expect uncertainty-aware models to become less certain for more severe perturbations. Posterior Network exhibits, as desired, the largest decrease in confidence with stronger corruptions (see Fig. 4b) while maintaining a high accuracy (see Fig. 4c).

| | Acc. | Alea. Conf. | Epist. Conf. | Brier | OOD Alea. | OOD Epist. | OODom Alea. | OODom Epist. |
|---|---|---|---|---|---|---|---|---|
| **PostNet: Conv.** | 78.58±0.1 | 95.45±0.0 | 93.36±0.0 | 33.84±0.2 | 72.21±0.1 | 57.72±0.7 | 100.0±0.0 | 100.0±0.0 |
| **PostNet: Alexnet** | 80.81±0.2 | 96.33±0.1 | 95.35±0.1 | 29.99±0.3 | 73.4±0.7 | 67.05±0.6 | 97.64±0.4 | 99.64±0.1 |
| **PostNet: VGG** | 84.85±0.0 | 97.76±0.0 | 97.25±0.0 | 22.84±0.0 | 80.21±0.2 | 77.71±0.3 | 91.35±0.5 | 99.25±0.1 |
| **PostNet: Resnet** | 87.86±0.2 | 98.35±0.0 | 97.13±0.0 | 19.33±0.3 | 79.92±0.4 | 72.25±0.6 | 99.94±0.0 | 99.94±0.0 |

Table 5: Results of PostNet with different encoder architectures. It shows good uncertainty estimation regardless of the architecture complexity.

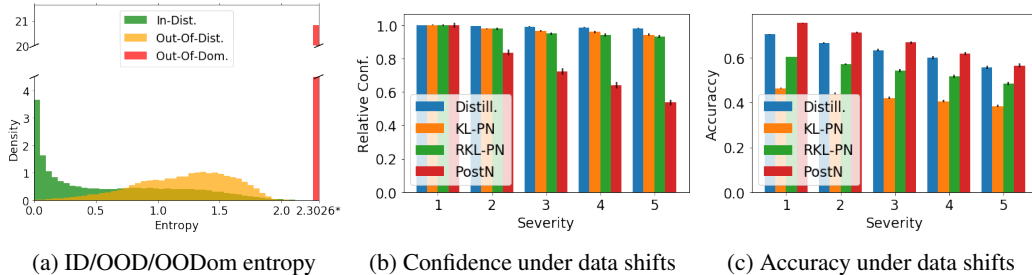

(a) ID/OOD/OODom entropy     (b) Confidence under data shifts     (c) Accuracy under data shifts

Figure 4: (a) shows entropy of the aleatoric distributions predicted by PostNet on CIFAR10 (ID) and SVHN (OOD, OODom). The value 2.3026* denotes the highest achievable entropy for 10 classes. PostNet can easily distinguish between the three data types. (b) and (c) present averaged confidence and accuracy under 15 dataset shifts introduced by [13] on CIFAR10 with conv. architecture. On more severe perturbations (i.e. data further away from data distribution), PostNet assigns higher epistemic uncertainty as desired. Baselines keeps same confidence even for less accurate predictions.

## 5 Conclusion

We propose Posterior Network, a model for uncertainty estimation in classification without requiring out-of-distribution samples for training or costly sampling for uncertainty estimation. PostNet is composed of three main components: an encoder which outputs a position in a latent space, a normalizing flow which performs a density estimation in this latent space, and a Bayesian loss for uncertainty-aware training. In our extensive experimental evaluation, PostNet achieves state-of-the-art performance with a strong improvement for detection of in- and out-of-distribution samples.

## Broader Impact

Traditional classification models without uncertainty estimate can be dangerous when used without domain expertise. They might have indeed unexpected behavior in new anomalous situations/input and are unaware of the underlying risk of their predictions. Uncertainty aware models like PostNet try to mitigate the risk of such autonomous predictions by attaching a confidence score to their predictions. On one hand, uncertainty aware predictions could be particularly beneficial in domains with potential critical consequences and prone to automation (e.g. finance, autonomous driving or medicine). When applied, these models are able to refrain from predicting if the data is out of their domain of expertise. Posterior Network makes a significant step further in this direction by even not requiring to observe similar anomalous situations during training. Anomalous data are typically not known in advance since they are rare by definition. Thus Posterior Network significantly increases the applicability of uncertainty estimation across application domains. On the other hand, high-quality of uncertainty estimation might also give a false sense of security. A potential risk is that an excessive trust in the model behavior leads to a lack of human supervision. For example in medicine, predictions wrongly deemed safe could have dramatic repercussions without human control.

## Acknowledgments and Disclosure of Funding

This research was supported by the BMW AG. The authors would like to thank Bernhard Schlegel for helpful discussion and comments.

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
