[Supplementary Material]

# A  Dirichlet Distribution Computations

## A.1  Dirichlet distribution

The Dirichlet distribution with concentration parameters $\boldsymbol{\alpha} = (\alpha_1, \dots, \alpha_C)$, where $\alpha_c > 0$, has the probability density function:

$$f(\boldsymbol{x}; \boldsymbol{\alpha}) = \frac{\prod_{c=1}^{C} \Gamma(\alpha_c)}{\Gamma\left(\sum_{c=1}^{C} \alpha_c\right)} \prod_{c=1}^{C} x_i^{\alpha_c - 1} \tag{8}$$

where $\Gamma$ is a gamma function:

$$\Gamma(\alpha) = \int_0^\infty \alpha^{z-1} e^{-\alpha} dz$$

## A.2  Closed-form formula for Bayesian loss.

The novel Bayesian loss described in formula 7 can be computed in closed form. For the sample $\mathbf{x}^{(i)}$, it is given by:

$$\mathcal{L}^{(i)} = \underbrace{\mathbb{E}_{q(\mathbf{p}^{(i)})}[\text{CE}(\mathbf{p}^{(i)}, \mathbf{y}^{(i)})]}_{(1)} - \underbrace{H(q^{(i)})}_{(2)} \tag{9}$$

where the distribution $q$ belongs to the family of the Dirichlet distributions $\text{Dir}(\boldsymbol{\alpha}^{(i)})$. The term (1) is the UCE loss [1]. Given that the observed class one-hot encoded by $\mathbf{y}^{(i)}$ is denoted by $c*$, the term (1) is equal to:

$$\mathbb{E}_{q(\mathbf{p}^{(i)})}[\text{CE}(\mathbf{p}^{(i)}, \mathbf{y}^{(i)})] = \Psi(\alpha_{c*}^{(i)}) - \Psi(\alpha_0^{(i)}) \tag{10}$$

where $\Psi$ denotes the digamma function. The term (2) is the entropy of a Dirichlet distribution and is given by:

$$H(q^{(i)}) = \log B(\boldsymbol{\alpha}^{(i)}) + (\alpha_0^{(i)} - C)\Psi(\alpha_0^{(i)}) - \sum_c (\alpha_c^{(i)} - 1)\Psi(\alpha_c^{(i)}) \tag{11}$$

where $B$ denotes the beta function.

## A.3  Epistemic covariance for in-distribution samples in PostNet.

The epistemic distribution in PostNet is a Dirichlet distribution $\text{Dir}(\boldsymbol{\alpha}^{(i)})$ with the following concentration parameters $\alpha_c^{(i)} = \beta_c^{\text{prior}} + N \cdot \mathbb{P}(c|\mathbf{z}^{(i)}; \phi) \cdot \mathbb{P}(\mathbf{z}^{(i)}; \phi)$ and $\alpha_0^{(i)} = \sum_c \beta_c^{\text{prior}} + N \cdot \mathbb{P}(\mathbf{z}^{(i)}; \phi)$. We can write the variance:

$$\text{Var}_{\mathbf{p} \sim \text{Dir}(\boldsymbol{\alpha}^{(i)})}(p_c) = \frac{\alpha_c(\alpha_0 - \alpha_c)}{\alpha_0^2(\alpha_0 + 1)}; \; \text{Cov}_{\mathbf{p} \sim \text{Dir}(\boldsymbol{\alpha}^{(i)})}(p_c, p_{c'}) = \frac{-\alpha_c \alpha_{c'}}{\alpha_0^2(\alpha_0 + 1)} \tag{12}$$

For in-distribution data (i.e. $\mathbb{P}(\mathbf{z}^{(i)}; \phi) \to \infty$), we have $\text{Var}_{\mathbf{p} \sim \text{Dir}(\boldsymbol{\alpha}^{(i)})}(p_c) = \mathcal{O}\left(\frac{\mathbb{P}(c|\mathbf{z}^{(i)}; \phi)(1 - \mathbb{P}(c|\mathbf{z}^{(i)}; \phi))}{\mathbb{P}(\mathbf{z}^{(i)}; \phi)N}\right)$ and $\text{Cov}_{\mathbf{p} \sim \text{Dir}(\boldsymbol{\alpha}^{(i)})}(p_c, p_{c'}) = \mathcal{O}\left(\frac{-\mathbb{P}(c|\mathbf{z}^{(i)}; \phi)\mathbb{P}(c'|\mathbf{z}^{(i)}; \phi)}{\mathbb{P}(\mathbf{z}^{(i)}; \phi)N}\right)$. Hence both terms converge to 0 when $\mathbb{P}(\mathbf{z}^{(i)}; \phi) \to \infty$.

# B  Model details

For vector datasets, all models share an architecture of 3 linear layers with Relu activation. A grid search in $[32, 64, 128]$ led to no significant changes in the performances. Therefore, we decided to use 64 hidden units for each layer. For image datasets, we used LeakyRelu activation and add on the top 3 convolutional layers with kernel size of 5, followed by a Max-pooling of size 2. Alternatively, we used the VGG16 architecture with batch normalization [37] adapted from PyTorch implementation [32]. All models are trained after a grid search for learning rate in $[1e^{-3}, 1e^{-5}]$. All models were optimized with Adam optimizer without further learning rate scheduling. We performed early stopping by

checking loss improvement every 2 epochs and a patience equal to 10. We trained all models on GPUs (1TB SSD).

For the dropout models, we used $p_{\text{drop}} = .25$ after a grid search in $[.25, .5, .75]$ and sampled 10 times for uncertainty estimation. As an indicator, the original paper [8], uses a dropout probability of .5 for MNIST. It also states that 10 samples already lead to reasonable uncertainty estimates. For the ensemble models, we used $m = 10$ networks after a grid search in $[2, 5, 10, 20]$. A greater number of networks was also found to give no great improvements in the original paper [21]. To be fair with these models, we distilled the knowledge of 10 neural networks for Distribution Distillation. We also trained Prior Networks where target parameters $\beta_{\text{in}} = 1e^2$ as suggested in original papers [24, 25].

For PostNet, we used a 1D batch normalization after the encoder. Experiments on latent dimensions and density types are presented in following sections. If not explicitley mentioned otherwise, we used by default radial flow with a flow length of 6 and a latent dimension of 6. This leads to only 80 parameters. Comparison with IAF are done with two layers of size 256. In general, we found out that a latent dimension smaller or equal to the number of classes is sufficient. It enables classes to be orthogonal in the latent space if necessary.

All metrics have been scaled by 100. We obtain numbers in $[0, 100]$ for all scores instead of $[0, 1]$.

## C   Datasets details

For all datasets, we use 5 different random splits to train all models. We split the data in training, validation and test sets ($60\%$, $20\%$, $20\%$). In particular, we did not restrict to classic MNIST and CIFAR10 splits in order do prevent overfitting to a specific split.

We use one toy dataset composed of three 2D isotropic Gaussians corresponding to three classes. The Gaussians means are $[0, 2.]$, $[-1.73205081, -1.]$ and $[1.73205081, -1.]$. The variance of the Gaussians is 0.2. A visualization of the true distributions for the three Gaussians is given in Figure 5. The final dataset is composed in total of 1500 samples.

We use the segment vector dataset [6], where the goal is to classify areas of images into 7 classes (window, foliage, grass, brickface, path, cement, sky). We remove the class 'sky' from training and instead use it as the OOD dataset for OOD detection experiments. Each input is composed of 18 attributes describing the image area. The dataset contains $2, 310$ samples in total.

We further use the Sensorless Drive vector dataset [6], where the goal is to classify extracted motor current measurements into 11 different classes. We remove classes 10 and 11 from training and use them as the OOD dataset for OOD detection experiments. Each input is composed of 49 attributes describing motor behaviour. The dataset contains $58, 509$ samples in total.

Additionally, we use the MNIST image dataset [22] where the goal is to classify pictures of hand-drawn digits into 10 classes (from digit 0 to digit 9). Each input is composed of a $1 \times 28 \times 28$ tensor. The dataset contains $70, 000$ samples. For OOD detection experiments, we use KMNIST [5] and FashionMNIST [40] containing images of Japanese characters and images of clothes, respectively.

Finally, we use the CIFAR10 image dataset [19] where the goal is to classify a picture of objects into 10 classes (airplane, automobile, bird, cat, deer, dog, frog, horse, ship, truck). Each input is a $3 \times 32 \times 32$ tensor. The dataset contains $60, 000$ samples. For OOD detection experiments, we use street view house numbers (SVHN) [29] containing images of numbers. For the dataset shift experiments, we use the classic split of CIFAR10 to avoid data leakage with the corrupted images from the test set that is provided online.

## D   Additional Experimental Results

In this section, we present additional results for uncertainty estimation on other datasets. Tables 6, 8, 9, 10, 12 show the performance of all models on the Segment, Sensorless Drive, MNIST, and CIFAR10 datasets. In the same way as for the other datasets, PostNet is competitive for all metrics and show a significant improvement on calibration among Dirichlet parametrized models and on OOD detection tasks among all models. We evaluated the performances of all models on MNIST using different uncertainty measures and observed very correlated results (see Table 11). We compared different encoder architectures on CIFAR10 (see Fig. 5). Without further parameter tuning,

PostNet adapted well to the convolutional architecture, AlexNet [20], VGG [37] and ResNet [12]. For easier comparison, we also trained models on the classic CIFAR10 split (79%, 5%, 16%) with VGG architecture. We noticed that a larger training set leads to better accuracy for all models.

We also show results of experiments with different latent dimensions (see Fig. 7; 8; 13; 14) and density types (MoG, radial, IAF) (see Tab. 6; 8; 9; 10; 12) for all datasets. We remarked that PostNet works with various type of densities even if using mixture of Gaussians presented more instability in practice. We observed no clear winner between Radial flow and IAF. We observed a bit lower performances for MoG which could be explained by its lack of expressiveness. Furthermore, we observed that a too high latent dimension would affect the performance.

Beside tables and figures with detailed metrics, we report additional visualizations. We present the uncertainty visualization on the input space for a 2D toy dataset (see Fig. 6). We show this visualization for all models parametrizing Dirichlet distributions. PostNet is the only model which is not overconfident for OOD data. In particular, it demonstrates the best fit of the true in-distribution data shown in figure 5. Other models show overconfident prediction for OOD regions and fail even on this simple dataset.

Furthermore, we plotted histograms of entropy of ID, OOD, OODom data for MNIST and CIFAR10 (see Fig 10 and 4). For both datasets, PostNet can easily distinguish between the three data types.

Finally, We also included the evolution of the uncertainty while interpolating linearly between images of MNIST (see Fig. 11 and 12). It corresponds to a smooth walk in latent space. As shown in Figure 11, PostNet predicts correctly on clean images and outputs more balanced class predictions for mixed images. Additionally the Figure 12 shows the evolution of the concentration parameters and consequently the epistemic uncertainty. We observe that the epistemic uncertainty (i.e. low $\alpha_c$) is higher on mixed images which do not correspond to proper digits.

Figure 5: Three Gaussians toy dataset.

| | Acc. | Alea. Conf. | Epist. Conf. | Brier | OOD Alea. | OOD Epist. |
|---|---|---|---|---|---|---|
| **Drop Out** | 95.25±0.1 | 99.75±0.0 | 99.43±0.0 | 11.89±0.2 | 41.48±0.5 | 43.11±0.6 |
| **Ensemble** | *97.27±0.1 | *99.88±0.0 | *99.85±0.0 | *7.64±0.2 | 54.76±1.6 | 58.13±1.7 |
| **Distill.** | 96.21±0.1 | 99.82±0.0 | **99.8±0.0** | 57.77±0.6 | 37.12±0.5 | 35.83±0.4 |
| **KL-PN** | 95.61±0.1 | 99.79±0.0 | 99.76±0.0 | 16.84±0.3 | 65.62±2.4 | 57.07±3.7 |
| **RKL-PN** | 96.36±0.2 | 99.71±0.0 | 99.58±0.0 | 11.97±0.1 | 75.46±2.4 | 51.02±0.6 |
| **PostN Rad. (2)** | 95.76±0.1 | 99.23±0.1 | 98.82±0.1 | 13.33±1.3 | 92.75±1.3 | 90.41±1.5 |
| **PostN Rad. (6)** | 96.52±0.2 | 99.82±0.0 | 99.43±0.0 | 8.69±0.3 | 98.27±0.2 | 98.09±0.3 |
| **PostN Rad. (10)** | 94.9±0.2 | 99.51±0.0 | 98.57±0.1 | 12.22±0.7 | 95.53±0.8 | 97.51±0.7 |
| **PostN IAF (2)** | 93.94±0.3 | 99.02±0.1 | 98.3±0.2 | 15.33±0.7 | *98.3±0.3 | *99.33±0.1 |
| **PostN IAF (6)** | 95.71±0.2 | 99.63±0.0 | 99.11±0.1 | 10.16±0.3 | 96.92±0.9 | 98.17±0.6 |
| **PostN IAF (10)** | **96.92±0.1** | **99.83±0.0** | 99.49±0.0 | **8.45±0.4** | 95.75±1.1 | 96.74±0.9 |
| **PostN MoG (2)** | 63.43±5.3 | 79.61±6.2 | 79.05±6.1 | 54.14±5.4 | 90.87±1.4 | 91.62±1.4 |
| **PostN MoG (6)** | 89.75±2.5 | 95.28±1.6 | 93.15±2.1 | 24.42±4.3 | 96.04±1.3 | 97.71±0.8 |
| **PostN MoG (10)** | 94.44±0.5 | 99.64±0.1 | 99.08±0.2 | 14.79±1.3 | 91.14±1.5 | 90.82±1.3 |

Table 6: Results on Segment dataset with all models. It shows results with different density types. Number into parentheses indicates flow size (for radial flow and IAF) or number of components (for MoG). Bold numbers indicate best score among Dirichlet parametrized models and starred numbers indicate best scores among all models.

Figure 6: Visualization of the concentration parameters predicted by Distribution Distillation, Prior Networks trained with KL and reverse KL divergence and Posterior Network on a 3-Gaussians toy dataset over 5 runs. Red dots indicate the mean of the 3 Gaussians. Colours indicate class labels predicted by the models, dark regions correspond to high epistemic uncertainty. PostNet consistently predicts low uncertainty around the training data and high uncertainty for OOD data.

| | Acc. | Alea. Conf. | Epist. Conf. | Brier | OOD Alea. | OOD Epist. |
|---|---|---|---|---|---|---|
| **PostN: No-Flow** | 93.13±0.3 | 99.48±0.1 | 98.41±0.3 | 12.94±0.3 | 47.3±2.9 | 35.49±0.3 |
| **PostN: No-Bayes-Loss** | 93.94±0.8 | 98.53±0.3 | 96.08±1.1 | 16.15±1.9 | 94.71±1.0 | 95.92±0.8 |
| **PostN: Seq-No-Bn** | 18.94±1.1 | 20.42±1.7 | 20.42±1.7 | 91.29±0.0 | 58.91±0.8 | 58.43±0.8 |
| **PostN: Seq-Bn** | 93.89±0.1 | 99.38±0.1 | 98.93±0.0 | 14.64±0.3 | 98.02±0.4 | 99.93±0.0 |

Table 7: Ablation study results on Segment dataset. Gray cells indicate significant drops in scores compare to the complete PostNet Rad. (6) model in Table 6.

Figure 7: Accuracy and uncertainty scores of PostNet with latent dimension in $[2, 6, 10, 32]$ on the Segment dataset. We observed that the performances remains high for small dimensions (i.e. $2$, $6$, $10$) and drop for a too high dimension (i.e. $32$).

| | Acc. | Alea. Conf. | Epist. Conf. | Brier | OOD Alea. | OOD Epist. |
|---|---|---|---|---|---|---|
| **Drop Out** | 89.32±0.2 | 98.21±0.1 | 95.24±0.2 | 28.86±0.4 | 35.41±0.4 | 40.61±0.7 |
| **Ensemble** | 99.37±0.0 | 99.99±0.0 | *99.98±0.0 | 2.47±0.1 | 50.01±0.0 | 50.62±0.1 |
| **Distill.** | 93.66±1.5 | 98.29±0.5 | 98.15±0.5 | 44.94±1.4 | 32.1±0.6 | 31.17±0.2 |
| **KL-PN** | 94.77±0.9 | 99.52±0.1 | 99.47±0.1 | 21.47±1.9 | 35.48±0.8 | 33.2±0.6 |
| **RKL-PN** | 99.42±0.0 | 99.96±0.0 | 99.89±0.0 | 9.07±0.1 | 45.89±1.6 | 38.14±0.8 |
| **PostN Rad. (2)** | 96.07±0.0 | 99.28±0.0 | 98.88±0.0 | 19.94±0.0 | *98.22±0.0 | *98.03±0.0 |
| **PostN Rad. (6)** | 98.02±0.1 | 99.89±0.0 | 99.47±0.0 | 5.51±0.2 | 72.89±0.8 | 88.73±0.5 |
| **PostN Rad. (10)** | 97.3±0.0 | 99.82±0.0 | 99.31±0.0 | 7.93±0.0 | 66.65±0.0 | 87.91±0.0 |
| **PostN IAF (2)** | 99.19±0.0 | 99.98±0.0 | 99.78±0.0 | 2.45±0.0 | 78.13±0.0 | 85.9±0.0 |
| **PostN IAF (6)** | 99.11±0.1 | 99.98±0.0 | 99.72±0.0 | 2.71±0.1 | 78.48±0.7 | 86.47±0.5 |
| **PostN IAF (10)** | *99.52±0.0 | *100.0±0.0 | 99.92±0.0 | *1.43±0.1 | 82.96±0.8 | 88.65±0.4 |
| **PostN MoG (2)** | 59.63±4.8 | 72.2±4.7 | 70.38±4.8 | 68.41±4.6 | 67.2±3.1 | 72.3±2.9 |
| **PostN MoG (6)** | 96.83±0.2 | 99.72±0.0 | 99.16±0.1 | 13.24±1.0 | 59.82±2.3 | 60.61±2.6 |
| **PostN MoG (10)** | 96.65±0.2 | 99.64±0.0 | 99.12±0.1 | 13.12±1.1 | 61.54±1.8 | 65.35±2.0 |

Table 8: Results on Sensorless Drive dataset with all models. It shows results with different density types. Number into parentheses indicates flow size (for radial flow and IAF) or number of components (for MoG).

Figure 8: Accuracy and uncertainty scores of PostNet with latent dimension in $[2, 6, 10, 32]$ on the Sensorless Drive dataset. OOD scores are computed against the left out sky class. We observed that the performances remains high for medium dimensions (i.e. 6, 10) and drop for a too high dimension (i.e. 32).

(a) ID/OOD data (PriorNet)

(b) ID/OOD data (PostNet)

Figure 9: This figure should be seen in perspective with Fig. 1. We plot FashionMNIST OODom data with black crosses to show where these data would land. OODom data were not used for training the models, A comparison of Fig. 9(a) with Fig. 1(b) show that Prior Network assigns high certainty to OODom data. In contrast, a comparison of Fig. 9(b) and Fig. 1(c) shows that Posterior Network assigns low uncertainty to OODom data as desired.

(a) MNIST

Figure 10: Histograms of the entropy of the predicted categorical distributions for in-distribution (green), out-of-distribution (yellow) and out-of-domain (red) data. The value $2.3026^*$ denotes the maximal entropy achievable for a categorical distribution with 10 classes. We use MNIST, FashionMNIST and the unscaled version of FashionMNIST as in-distribution, out-of-distribution and out-of-domain data. PostNet clearly distinguishes between the three types of data with low entropy for in-distribution data and high entropy for out-of-distribution, and close to the maximum possible entropy for out-of-domain data.

Figure 11: Evolution of the probability predictions when interpolating linearly between four MNIST images. The interpolation goes from the clean digits 5, 4, 6 and 2 in a cyclic way with 20 interpolated images between each pair. As desired, We can observe correct predictions around clean images with higher (aleatoric) uncertainty for mixed images, and smooth transitions in between.

Figure 12: Evolution of the concentration parameters predictions when interpolating linearly between four MNIST images. The interpolation goes from the clean digits 5, 4, 6 and 2 in a cyclic way with 20 interpolated images between each pair. As desired, we can observe correct and confident predictions around clean images with higher (epistemic) uncertainty for mixed images.

|  | Acc. | Alea. Conf. | Epist. Conf. | Brier |
|---|---|---|---|---|
| **Drop Out** | 99.26±0.0 | 99.98±0.0 | 99.97±0.0 | 1.78±0.0 |
| **Ensemble** | *99.35±0.0 | *99.99±0.0 | *99.98±0.0 | 1.67±0.0 |
| **Distill.** | **99.34±0.0** | **99.98±0.0** | *99.98±0.0 | 72.55±0.2 |
| **KL-PN** | 99.01±0.0 | 99.92±0.0 | 99.95±0.0 | 10.82±0.0 |
| **RKL-PN** | 99.21±0.0 | 99.67±0.0 | 99.57±0.0 | 9.76±0.0 |
| **RKL-PN w/ F.** | 99.2±0.0 | 99.75±0.0 | 99.68±0.0 | 9.9±0.0 |
| **PostN Rad. (2)** | **99.34±0.0** | **99.98±0.0** | 99.97±0.0 | *1.25±0.0 |
| **PostN Rad. (6)** | 99.28±0.0 | 99.97±0.0 | 99.96±0.0 | 1.36±0.0 |
| **PostN Rad. (10)** | 99.22±0.0 | 99.97±0.0 | 99.97±0.0 | 1.41±0.0 |
| **PostN IAF (2)** | 99.06±0.0 | 99.96±0.0 | 99.94±0.0 | 1.48±0.0 |
| **PostN IAF (6)** | 99.08±0.0 | 99.96±0.0 | 99.94±0.0 | 1.45±0.1 |
| **PostN IAF (10)** | 98.97±0.0 | 99.96±0.0 | 99.94±0.0 | 1.61±0.0 |
| **PostN MoG (2)** | 76.41±2.3 | 99.93±0.0 | 99.92±0.0 | 23.23±2.2 |
| **PostN MoG (6)** | 99.21±0.0 | 99.94±0.0 | 99.92±0.0 | 1.61±0.0 |
| **PostN MoG (10)** | 99.22±0.0 | 99.94±0.0 | 99.92±0.0 | 1.53±0.0 |

Table 9: Accuracy, confidence and calibration results on MNIST dataset with all models. It shows results with different density types. Number into parentheses indicates flow size (for radial flow and IAF) or number of components (for MoG). Bold numbers indicate best score among Dirichlet parametrized models and starred numbers indicate best scores among all models.

|  | OOD K. Alea. | OOD K. Epist. | OOD F. Alea. | OOD F. Epist. | OODom K. Alea. | OODom K. Epist. | OODom F. Alea. | OODom F. Epist. |
|---|---|---|---|---|---|---|---|---|
| **Drop Out** | 94.0±0.1 | 93.01±0.2 | 96.56±0.2 | 95.0±0.2 | 31.59±0.5 | 31.97±0.5 | 27.2±1.1 | 27.52±1.1 |
| **Ensemble** | *97.12±0.0 | *96.5±0.0 | 98.15±0.1 | 96.76±0.0 | 41.7±0.3 | 42.25±0.3 | 37.22±1.0 | 37.73±1.0 |
| **Distill.** | **96.64±0.1** | 85.17±1.0 | 98.83±0.0 | 94.09±0.4 | 11.49±0.3 | 10.66±0.2 | 13.82±0.5 | 12.03±0.3 |
| **KL-PN** | 92.97±1.2 | 93.39±1.0 | 98.44±0.1 | 98.16±0.0 | 9.54±0.1 | 9.78±0.1 | 9.57±0.1 | 10.06±0.1 |
| **RKL-PN** | 60.76±2.9 | 53.76±3.4 | 78.45±3.1 | 72.18±3.6 | 9.35±0.1 | 8.94±0.0 | 9.53±0.1 | 8.96±0.0 |
| **RKL-PN w/ F.** | 81.34±4.5 | 78.07±4.8 | *100.0±0.0 | *100.0±0.0 | 9.24±0.1 | 9.08±0.1 | 88.96±4.4 | 87.49±5.0 |
| **PostN Rad. (2)** | 95.49±0.3 | 93.12±0.7 | 96.2±0.3 | 94.6±0.4 | *100.0±0.0 | *100.0±0.0 | *100.0±0.0 | *100.0±0.0 |
| **PostN Rad. (6)** | 95.75±0.2 | **94.59±0.3** | 97.78±0.2 | 97.24±0.3 | *100.0±0.0 | *100.0±0.0 | *100.0±0.0 | *100.0±0.0 |
| **PostN Rad. (10)** | 95.46±0.4 | 94.19±0.4 | 97.33±0.2 | 96.75±0.3 | *100.0±0.0 | *100.0±0.0 | *100.0±0.0 | *100.0±0.0 |
| **PostN IAF (2)** | 92.24±0.3 | 91.75±0.3 | 96.58±0.2 | 96.6±0.2 | *100.0±0.0 | *100.0±0.0 | *100.0±0.0 | *100.0±0.0 |
| **PostN IAF (6)** | 90.74±0.6 | 90.63±0.6 | 93.66±0.5 | 93.17±0.6 | *100.0±0.0 | *100.0±0.0 | *100.0±0.0 | *100.0±0.0 |
| **PostN IAF (10)** | 87.08±0.2 | 86.52±0.3 | 92.34±0.6 | 91.27±0.9 | *100.0±0.0 | *100.0±0.0 | *100.0±0.0 | *100.0±0.0 |
| **PostN MoG (2)** | 74.27±2.0 | 73.34±1.9 | 76.99±2.0 | 76.74±1.9 | *100.0±0.0 | *100.0±0.0 | 99.99±0.0 | 99.99±0.0 |
| **PostN MoG (6)** | 84.67±1.5 | 81.46±1.9 | 88.98±1.7 | 87.07±2.1 | *100.0±0.0 | *100.0±0.0 | *100.0±0.0 | *100.0±0.0 |
| **PostN MoG (10)** | 85.14±1.3 | 81.12±1.5 | 94.43±0.8 | 93.8±1.0 | *100.0±0.0 | *100.0±0.0 | *100.0±0.0 | *100.0±0.0 |

Table 10: OOD results on MNIST dataset with all models. It shows results with different density types. Number into parentheses indicates flow size (for radial flow and IAF) or number of components (for MoG).

|  | OOD K. $\alpha_0$/var. | OOD K. MI. | OOD F. $\alpha_0$/var. | OOD F. MI. | OODom K. $\alpha_0$/var. | OODom K. MI. | OODom F. $\alpha_0$/var. | OODom F. MI |
|---|---|---|---|---|---|---|---|---|
| **Ensemble** | *97.19±0.0 | *97.44±0.0 | 97.53±0.1 | 97.69±0.1 | 42.36±0.3 | 42.38±0.3 | 37.85±1.1 | 37.86±1.1 |
| **RKL-PN** | 54.11±3.4 | 54.9±3.3 | 72.54±3.6 | 73.33±3.5 | 8.94±0.0 | 8.94±0.0 | 8.96±0.0 | 8.96±0.0 |
| **RKL-PN w/ F.** | 78.4±4.8 | 78.73±4.8 | *100.0±0.0 | *100.0±0.0 | 9.08±0.1 | 9.08±0.1 | 87.49±5.0 | 87.49±5.0 |
| **PostN** | **96.04±0.2** | **96.05±0.2** | 98.17±0.2 | 98.17±0.2 | *100.0±0.0 | *100.0±0.0 | *100.0±0.0 | *100.0±0.0 |

Table 11: OOD detection on MNIST with other uncertainty measures. Mutual Information [24] and $\alpha_0$ (Dirichlet) / variance (Ensemble) results are highly correlated.

Figure 13: Accuracy and uncertainty scores of PostNet with latent dimension in $[2, 6, 10, 32]$ on the MNIST dataset. OOD and OODom scores are computed against scaled and unscaled KMNIST and FashionMNIST datasets. We observed that the performances remains high for medium dimensions (i.e. 6, 10) and drop for a too high dimension (i.e. 32).

| | Acc. | Alea. Conf. | Epist. Conf. | Brier | OOD Alea. | OOD Epist. | OODom Alea. | OODom Epist. |
|---|---|---|---|---|---|---|---|---|
| **Drop Out** | 71.73±0.2 | 92.18±0.1 | 84.38±0.3 | 49.76±0.2 | 72.94±0.3 | 41.68±0.5 | 28.3±1.8 | 47.1±3.3 |
| **Ensemble** | *81.24±0.1 | *96.61±0.0 | 93.25±0.1 | 38.88±0.1 | *77.82±0.2 | 55.17±0.3 | 63.18±1.1 | 89.97±0.9 |
| **Distill.** | 72.11±0.4 | 91.72±0.2 | 90.73±0.2 | 88.04±0.1 | **75.63±0.6** | 52.18±2.1 | 17.76±0.0 | 17.76±0.0 |
| **KL-PN** | 48.84±0.5 | 78.01±0.6 | 77.99±0.7 | 83.11±0.6 | 59.32±1.1 | 58.03±0.8 | 17.79±0.0 | 20.25±0.2 |
| **RKL-PN** | 62.91±0.3 | 85.62±0.2 | 81.73±0.2 | 58.12±0.4 | 67.07±0.4 | 56.64±0.8 | 17.83±0.0 | 17.76±0.0 |
| **PostN Rad. (2)** | 76.43±0.1 | 94.59±0.1 | 94.02±0.1 | 37.59±0.3 | 72.91±0.4 | 69.26±1.1 | 99.99±0.0 | *100.0±0.0 |
| **PostN Rad. (6)** | 76.46±0.3 | 94.75±0.1 | *94.34±0.1 | *37.39±0.4 | 72.83±0.6 | *72.82±0.7 | *100.0±0.0 | *100.0±0.0 |
| **PostN Rad. (10)** | 75.43±0.2 | 94.16±0.1 | 93.64±0.1 | 39.3±0.4 | 71.94±0.3 | 70.99±0.5 | *100.0±0.0 | *100.0±0.0 |
| **PostN IAF (2)** | 76.75±0.2 | **94.78±0.1** | 92.98±0.2 | 37.87±0.5 | 73.07±0.5 | 65.61±1.0 | *100.0±0.0 | *100.0±0.0 |
| **PostN IAF (6)** | **76.79±0.1** | 94.73±0.0 | 93.7±0.1 | 37.86±0.2 | 73.58±0.2 | 69.74±0.3 | *100.0±0.0 | *100.0±0.0 |
| **PostN IAF (10)** | 75.92±0.2 | 94.48±0.1 | 93.23±0.2 | 39.09±0.3 | 72.4±0.3 | 69.04±0.3 | *100.0±0.0 | *100.0±0.0 |
| **PostN MoG (2)** | 44.7±5.9 | 54.12±7.9 | 52.12±7.8 | 68.57±5.2 | 48.53±3.6 | 47.45±4.1 | 99.91±0.0 | 99.96±0.0 |
| **PostN MoG (6)** | 71.05±1.6 | 91.21±1.0 | 86.91±1.2 | 46.37±2.0 | 73.49±0.6 | 56.04±3.8 | 98.04±0.7 | 99.62±0.1 |
| **PostN MoG (10)** | 71.63±1.3 | 91.57±0.8 | 88.92±0.8 | 46.07±1.9 | 72.61±0.3 | 56.28±1.8 | 99.88±0.0 | **100.0±0.0** |

Table 12: Results on CIFAR10 dataset with all models with convolutional architecture. It shows results with different density types. Number into parentheses indicates flow size (for radial flow and IAF) or number of components (for MoG).

| | Acc. | Alea. Conf. | Epist. Conf. | Brier | OOD S. Alea. | OOD S. Epist. | OODom S. Alea. | OODom S. Epist. |
|---|---|---|---|---|---|---|---|---|
| **Ensemble** | *91.34±0.0 | *99.1±0.0 | 98.77±0.0 | 17.69±0.1 | *80.1±0.3 | 75.14±0.2 | 21.1±3.1 | 24.42±3.7 |
| **RKL-PN** | 60.05±0.7 | 85.63±0.8 | 82.11±1.3 | 70.84±0.9 | 50.97±3.9 | 55.37±4.3 | 56.16±1.4 | 51.33±2.4 |
| **RKL-PN w/ C100** | 88.18±0.1 | 95.44±0.3 | 94.15±0.3 | 79.99±2.0 | 56.67±2.1 | 73.37±2.3 | 57.06±1.7 | 50.31±1.4 |
| **PostNet** | **90.05±0.1** | **98.87±0.0** | *98.82±0.0 | *15.44±0.1 | 76.04±0.4 | *75.57±0.4 | *87.65±0.3 | *92.13±0.5 |

Table 13: Results with VGG16 on CIFAR10 on classic split (79%, 5%, 16%). RKL-PN w/ C100 uses CIFAR100 as training OOD.

Figure 14: Accuracy and uncertainty scores of PostNet with latent dimension in $[2, 6, 10, 32]$ on the CIFAR10 dataset. OOD and OODom scores are computed against scaled and unscaled SVHN dataset. We observed that the performances remains high for medium dimensions (i.e. 6, 10) and drop for a too high dimension (i.e. 32).