[Reviews · NeurIPS 2020]

Review 1

Summary and Contributions: The work proposes a solution to the two main limitations of Dirichlet Prior Networks - having to specify an arbitrary high precision in-domain and the need for OOD training data. In this work normalising flows over a latent-representation are used to jointly-learn and control the per-prediction concentration parameters of a Dirichlet Prior Network.

Strengths: I quite like the main idea of this paper. It is both a novel modification of Prior Networks, and relevant to the NeurIPS community. It is based on solid theoretical grounding (DPNs, Normalising flows) and proposes an elegant solution to the two main issues of DPNs. The proposed method is evaluated on two tabular datasets, and on the MNIST and CIFAR-10 datasets.

Weaknesses: While the paper proposes an interesting solution, I believe it falls short on a range of aspects which greatly affected my score. Theoretical. 1. It is not clear what measures of uncertainty are used for OOD detection. Previous work on Prior Networks and ensemble methods consistently make use of mutual information to obtain a separable set of estimates of total, aleatoric and epistemic uncertainty. However, this work does neither mentions this nor uses these *established* and *theoretically meaningful* measures. Rather perplexingly, this work seems to make use of max alpha_c^{I} scores for Prior Network and variance of probability for ensembles. While this may have an interpretation, it needs to be developed in far greater detail, rather than hidden away in the discussion of the experimental setup. Furthermore, It is interesting how the proposed method affects estimates of aleatoric/epistemic uncertainty separately. Finally, using the established decomposition based on mutual information also allows comparing ensembles and Prior Networks on equal terms. 2. The Uncertainty aware loss function - the author's fail to point out that this is exactly equivalent to the RKL loss defined by Malinin et al (2019) with a target beta = 1, and also equivalent to an ELBO loss. Thus, what the authors are in fact doing is maximising the ELBO with a uniform prior over the Dirichlet latent variable. This connection should really be highlighted to make a proper connect to prior work on VAEs and other latent variable models. 3. The author's claim that placing a Flow-based model over a latent representation does not suffer from the same problems as flow-based models over the image space needs to be further developed. 4. Is there a limit on scaling for the model? If a flow needs to be trained with very class, does this limit deployment to tasks with >100 classes? Empirical 1. The primary concern regarding the experiments is that the models achieve very poor classification accuracy on CIFAR10 using the VGG architecture. While by no means SOTA, VGG16 can nevertheless achieve over 92-93% accuracy on CIFAR10. However, the models presented here have, at best, an accuracy of ~84%. The creates two concerns. Firstly, the models may be either under-trained, and mis-specified in some way, which clearly will impact the quality of uncertainty estimates. More generally, is a method provides good uncertainty, but poor quality, it is unlikely to ever be used. However, considering that RKL-DPNs can achieve a far higher CIFAR-10 accuracy than presented in this paper, I believe it is more likely that the training procedure was mis-specified in someway. 2. The author's make a strange distinction between aleatoric OOD and epistemic OOD. Are different uncertainty measures being used? Also, why is AUC-PR used, rather tan AUC-ROC, as is standard for OOD detection? This makes it difficult to compare these number to prior work. 3. Thorough comparisons of PostNet and Ensembles on CIFAR-10 are missing. More generally, the method should be validated on a larger dataset (CIFAR-100, TinyImageNet, ImageNet). It is interesting how the current method scales with the number of classes. 4. A greater and more diverse range of OOD datasets needs to be considered. Distinguishing CIFAR-10 from SVHN is not particularly difficult. Until a proper comparison is made with results from prior work, it is difficult to accept the claim of SOTA OOD detection results.

Correctness: I believe the paper is generally sound. Concerns expressed above.

Clarity: The paper is could be better written. Crucially, the authors put far too much math in text, which makes reading difficult, and have a baroque notation, often referring to the same thing in different ways, which makes understanding difficult. I would suggest that the authors simplify their notations and move math out of the main body of text.

Relation to Prior Work: This work is in some sense related to the Mahalanobis classifier approach, which also does OOD detection by specifying a (simple) generative model of the final-layer representation of a classifier. The connection should be highlighted.

Reproducibility: Yes

Additional Feedback: --- POST REBUTTAL COMMENTS --- I believe that the authors have sufficiently addressed the issues I have raised to enable me to update my review to a 6. I really like the idea. However, while the authors have have a great idea, I strongly believe evaluation and discussion are lacking. Firstly, I would still like the authors to add a detailed discussion of the possible limitations of using flows even in the latent space. The rebuttal mostly re-iterated what was said in the paper, and did not really address my concerns. Specifically, the transformation may be such that multiple points in the input space are mapped to the same location in latent space. Thus, constructing a generative model in the latent space may not be able to fully capture all forms of anomalies in the input space. Nothing in the network so far addresses this issue. You could, for example, make the argument that the features extracted as part of the classification task enable OOD detection in the context of the discriminative task ("Do I know how to classify this?" vs "Is the input anomalous?" ). Secondly, the issue of scaling (computational and memory) to a large number of classes has not been addressed. Do we need a flow per class? Does this mean that for an image-net scale model we need to train 1000 flows? Is this a principle limitation of the proposed approach? Thirdly, OOD detection of MNIST and CIFAR10 vs SVHN is not a sufficient evaluation to claim SOTA OOD in 2020. It should not be difficult to evaluated CIFAR10 vs a range of other datasets (C100, TinyImageNet, LSUN, ImageNet-C, etc....). This is a simple and necessary extension. It would also be good to *train* a PostNet on CIFAR100/TIM, etc... to evaluate scaling to larger and more interesting datasets with greater numbers of classes (relates to previous point). This would take time. However, I strongly encourage the authors to do this for the camera-ready version of the paper. It would make the paper have far greater impact than as-is.


Review 2

Summary and Contributions: The authors proposed the use of normalizing flows combined with a dimensionality reducing encoder to learn the Dirichlet evidence without having to use out of distribution training examples. The paper includes extensive experiments to demonstrate the effectiveness of the approach such as well calibrated posterior predictive probabilities as well as high epistemic uncertainty for out of distributions test samples.

Strengths: The idea to treat the evidence as the learned likelihood for each class (via normalized flows) scaled by the amount of training examples per a class is really clever. I also like that the paper experimented with alternative density estimation techniques. The overall approach removes the requirement to use out of distribution samples that are needed by other methods that attempt to learn Dirichlet evidence such as prior networks. The experiments demonstrate the superior performance of the proposed posterior network method.

Weaknesses: The paper claims that evidential deep learning (EDL) [30] requires OOD training samples. That is not the case, but EDL in [30] does not exhibit the desirable properties shown in fig. 1 (d). An updated version of EDL was presented at AAAI 2020 that does exhibit similar desirable properties, and it also does not require one to provide OOD samples. Rather, it combines VAEs and GANs to generate out of distribution samples used to define evidence. Nevertheless, the proposed approach seems to be more eloquent. [Response to Author Feedback] I appreciate that the authors will update their characterization of the EDL. I believe that this paper would make a nice addition to the NeurIPS program. I also agree with the other reviewers that the empirical evaluation can be enhanced.

Correctness: The results presented in the paper appear to be correct.

Clarity: The paper is well written and is a joy to read.

Relation to Prior Work: For the most part, the paper sufficiently describes the prior work on Bayesian neural networks, approximation thereof, and other Dirichlet evidence approaches. There is a slight mischaracterization of EDL as described above and a recent update of EDL that was inadvertently missed.

Reproducibility: Yes

Additional Feedback: Of minor note: Line 80, I do think that alpha_c<1 is a problem in practice. It simply translates to high epistemic uncertainty. Line 245: Why is SVHN not an OOD dataset for MNIST? In the experimental section, it might not be best to treat the inverse variance as the epistemic uncertainty for the ensemble and drop-out methods. Rather, it might be more fair to use p_max(1-p_max)/var as this is more representative of the Dirichlet strength alpha_0. As a nitpick, the encoder approach in this paper might not be amenable to promoting adversarial robustness.


Review 3

Summary and Contributions: The paper proposes to combine neural networks with normalising flows to estimate the parameters of dirichlet posterior distributions. The proposed method improves over regular Dirichlet Prior Networks by not requiring any OOD data and (according to experiments) providing superior predictive performance with improved uncertainty estimation. The paper shows experiments on some toy data as well as MNISTlike datasets and CIFAR10 and same tabular data.

Strengths: The paper does a good job at introducing and motivation the use of normalising flows for the estimation of Dirichlet parameters to estimate more reliable Dirichlet parameters than with Dirichlet Prior Networks. The used normalising flows help to estimate the parameters of the Dirichlet while integrating the amount of seen training data. The paper is easy to follow and offers a novel and potentially impactful approach to uncertainty quantification with neural networks. The experiments confirm the performance of PostNet in terms of predictive accuracy as well as uncertainty estimation on a range of different tasks.

Weaknesses: The paper is generally very interesting to read and novel. However, the experimental section could use some improvement by integrating stronger baseline comparisons as well as digging deeper in the differences between different flow choices that have been presented in the supplement and seem to indicate that the choice in flows might not be as straight forward as claimed. The paper mentions that DPN always requires OOD data but then only uses random noise for training DPNs - which could be argued to not be OOD data but noise. It seems that this choice in training (together with potential other differences) leads to significantly subpar performances of the DPNs compared to their originally reported performances. It would be beneficial to comment on those disparities to allow for a more objective view on PostNet. Further, it seems that relevant comparisons e.g. with ensembles have purposely been hidden away in the supplement rather than showing them in Table 4.

Correctness: As far as I can tell, both the method and empirical methodology seem correct. However, the choice of OOD data for DPNs is questionable.

Clarity: The paper is well written and easy to follow. The motivation and intuition behind the method is well explained. My main problem with clarity are some formatting choices that seem to have been made because of page limit. It would be beneficial to expand the equations from inline into full equation displays to allow for better readability as some subscripts (e.g. c on line 248/249) intersect the line below.

Relation to Prior Work: The paper does a good job at differentiating its proposed method from previous contributions and compares most to most relevant works. Depending on the focus it could be relevant to cite works that use flows for OOD detection such as [1], however, this does not seem required as the current focus seems more on uncertainty estimation in general rather than OOD detection. [1] Zisselman, Ev, and Aviv Tamar. "Deep Residual Flow for Out of Distribution Detection." Proceedings of the IEEE/CVF Conference on Computer Vision and Pattern Recognition. 2020.

Reproducibility: Yes

Additional Feedback: Post-rebuttal: Thanks for addressing my comments. I have increased my score to 6. However, my score increase is strongly motivated by me liking the core idea of the paper and I encourage the authors to consider some of the other reviewers comment for improving the experimental evaluation and discussion. ======================= - L8: Some methods use OOD data but there are also a lot of methods not using OOD data. This statement seems to focussed on Dirichlet Prior Networks. - L 29: What are sub-models? It seems like it is talked about ensembles? - Sec 2.1 has a lot of inline equations that would be easier to read when displayed. This is also a problem later in the paper. - Table 2 should add the a row with the full proposed method for easier comparisons. - Table 4 shows results that are much worse than regular DPN papers. Why is this the case? Also the statement about requiring OOD data is somewhat unfair as you don't necessarily need specific OOD data and are only training using random noise (apart from the FashionMNIST example) rather than using structured OOD data. - On a more personal preference, I believe that Fig 3 would be easier on the eyes with a white background. - Fig 4: It would be useful to simply use log_10 which then gives [0, 1] values. - L 318: It mentions that PostNet achieves SoTA performance. However, it lacks comparisons to more recent works with better performance. - The supplement also shows results with ResNets - in general, I would recommend the authors to include ResNet results in the main paper for a potential camera reader version as that increases the trust of the applicability to current deep learning architectures. - How is the ablation study with no flows performed? How is the likelihood estimated?


Review 4

Summary and Contributions: Posterior Network (PostNet) is proposed in this paper to estimate uncertainty of in-and out-of-distribution data. Specifically, it uses normalizing flows to predict an individual closed=form posterior distribution over predicted probabilities for any input sample. Compared with previous works, PostNet does not require access to OOD data at training time.

Strengths: 1. Compared with previous works, PostNet does not require access to OOD data at training time. I believe the removal of this criteria makes the proposed method more general and can be easily applied to more applications. 2. Both qualitative and quantitative comparisons are provided. Results show that PostNet is superior than baseline models. 3. Ablation study is conducted to justify the effectiveness of each part of the proposed module.

Weaknesses: I think mostly the model is well presented and evaluated without significant issue. One thing I am curious is that how effective the proposed method is for dataset shift in addition to OOD. Specifically, it is a well known effect that there is dataset shift problem for a lot of applications, including the financial time-series application mentioned by the authors. I think the proposed method might be a new way to deal with this problem. It will be very interesting if the authors can provide some discussion on this.

Correctness: Yes.

Clarity: Yes, overall the paper is well written. Claims and conclusions are backed up by experiments and comparisons. The description on previous works and contributions are clear.

Relation to Prior Work: Yes. The description on previous works and contributions are clear.

Reproducibility: Yes

Additional Feedback: After the rebuttal, I believe the authors have sufficiently addressed my concerns. Therefore, I'll stay on the positive side to accept the paper.

[Author Response · NeurIPS 2020]

| | OOD K. $\alpha_0$/var. | OOD K. MI. | OOD F. $\alpha_0$/var. | OOD F. MI. | OODom K. $\alpha_0$/var. | OODom K. MI. | OODom F. $\alpha_0$/var. | OODom F. MI. |
|---|---|---|---|---|---|---|---|---|
| **Ensemble** | *97.19±0.0 | *97.44±0.0 | 97.53±0.1 | 97.69±0.1 | 42.36±0.3 | 42.38±0.3 | 37.85±1.1 | 37.86±1.1 |
| **RKL-PN** | 54.11±3.4 | 54.9±3.3 | 72.54±3.6 | 73.33±3.5 | 8.94±0.0 | 8.94±0.0 | 8.96±0.0 | 8.96±0.0 |
| **RKL-PN w/ F.** | 78.4±4.8 | 78.73±4.8 | *100.0±0.0 | *100.0±0.0 | 9.08±0.1 | 9.08±0.1 | 87.49±5.0 | 87.49±5.0 |
| **PostN** | 96.04±0.2 | 96.05±0.2 | 98.17±0.2 | 98.17±0.2 | *100.0±0.0 | *100.0±0.0 | *100.0±0.0 | *100.0±0.0 |

Table A: OOD detection (MNIST). MI and $\alpha_0$ (Dirichlet) / variance (Ensemble) results are highly correlated.

| | Acc. | Alea. Conf. | Epist. Conf. | Brier | OOD S. Alea. | OOD S. Epist. | OODom S. Alea. | OODom S. Epist. |
|---|---|---|---|---|---|---|---|---|
| **Ensemble** | *91.34±0.0 | *99.1±0.0 | 98.77±0.0 | 17.69±0.1 | *80.1±0.3 | 75.14±0.2 | 21.1±3.1 | 24.42±3.7 |
| **RKL-PN** | 60.05±0.7 | 85.63±0.8 | 82.11±1.3 | 70.84±0.9 | 50.97±3.9 | 55.37±4.3 | 56.16±1.4 | 51.33±2.4 |
| **RKL-PN w/ C100** | 88.18±0.1 | 95.44±0.3 | 94.15±0.3 | 79.99±2.0 | 56.67±2.1 | 73.37±2.3 | 57.06±1.7 | 50.31±1.4 |
| **PostNet** | 90.05±0.1 | 98.87±0.0 | *98.82±0.0 | *15.44±0.1 | 76.04±0.4 | *75.57±0.4 | *87.65±0.3 | *92.13±0.5 |

Table B: Results (VGG16) on CIFAR10 on classic split. RKL-PN w/ C100 uses CIFAR100 as training OOD.

**Uncertainty metrics (R1).** Based on R1's comments we also evaluated the models based on mutual
information (Tab. A). MI is highly correlated with both $\alpha_0$ and variance with barely score changes.

**AUROC vs APR (R1).** Both metrics have been used by prior works to assess OOD detection
performance [20,4,A,B]. Theoretically, the two metrics bring similar information [C]. In practice,
APR is preferred when working with imbalanced classes (such as anomaly detection) since AUROC
might lead to too optimistic results [D]. For these reasons, we decided to use APR.

**Flow on input vs latent space (R1).** As shown in existing work, distinguishing between CIFAR10
and SVHN is not trivial [4, 24, E]. We attribute the strong performance of PostNet to the dim.
reduction and the classification task (Sect. 2.2). Similar conclusions have been drawn in [E].

**PostNet CIFAR10 acc. (R1).** PostNet provides both good uncertainty estimates and accuracy (Fig. 5,
7, 8, 9, 10). In our paper we use 5 random splits (60%, 20%, 20%). Based on R1's comments,
we also trained on the classic split (79%, 5%, 16%). PostNet achieves ∼90% accuracy (Tab. B).
Experiments using random splits lead to better estimates of the true model performance. We made a
proper comparison focused on small number of classes similar to [20] and enrich experiments with
tabular, shifts and OODom settings. We agree that results on more classes are interesting future work.

**PriorNet acc. w/o OOD (R1, R3).** Specifying training OOD data is unrealistic (l.70-71) and is
unlikely to generalize to all other OOD datasets (l.72-74). We demonstrate these issues with practical
results (Tab. 3, Tab. B). Indeed, the results of PriorNets deteriorate when the OOD data used at training
time (e.g. noise/KMNIST/CIFAR100) differs from the OOD data at test time (e.g. FMNIST/SVHN).
Still, we also report results for PriorNet with true OOD on MNIST and CIFAR10, where it obtains
∼99% and ∼89% accuracy, respectively. This is similar to reported results in [21], ruling out the
possibility of under-trained or mis-specified models.

**Ensemble baseline (R1, R3).** We provide results of Ensemble in Fig. 5, 7, 8, 9, 10 in app. and
additionally on CIFAR10 with VGG16 in Tab. B. Ensemble has a high training cost which justified
a specific treatment. Note that Tab. 4 aims at comparing models training a single network (l.298),
this is why here ensemble is not included. Ensemble achieves good performance except for tabular
left-out classes and OODom datasets where PostNet shows substantially better results.

**Flow choice (R3).** In our experiments (app. Fig. 5, 7, 8, 9, 10), both flows (e.g. PostN Rad (6) and
PostN IAF (6)) achieve good performance on the four datasets, even though the flow depth can impact
the performance. Using MoG leads to weaker performance. Note that the No-Flow model outputs $\alpha$
which are directly used to compute the Bayesian loss (no likelihood with NF or MoG).

**Stronger baselines (R3).** We compare PostNet to recent Dirichlet-based SoTA methods (2018 and
newer). We also consider Drop-Out and Ensembles, which are strong baselines [20, 21, 33].

**Dataset shifts (R4).** Fig. 4 shows that PostNet assigns lower confidence to larger dataset shifts (l.264).

**Related work (R1, R2, R3).** We will include suggestions and correct the misleading EDL statement.
In particular, we will explain connections between RKL and the Bayesian loss.

[A] Hendrycks et al. "Deep Anomaly Detection with Outlier Exposure". ICLR 2019.
[B] Hendrycks et al. "A Baseline for Detecting Misclassified & OOD Examples in Neural Networks". ICLR 2017.
[C] Jesse et al. Davis. "The Relationship Between Precision-Recall and ROC Curves". ICML 2006.
[D] Takaya Saito and Marc Rehmsmeier. "The Precision-Recall Plot Is More Informative than the ROC Plot
When Evaluating Binary Classifiers on Imbalanced Datasets". PloS one 2015.
[E] Kirichenko et al. "Why Normalizing Flows Fail to Detect OOD Data". Arxiv 2020.



[Meta-Review · NeurIPS 2020]

The work has clever ideas with considerable advances on Prior Networks, and all reviewers found the paper interesting and well-written. There are notable limitations that the work should be honest about, however, and I strongly recommend that the authors make revisions following the reviews and rebuttal. For example: + There are a number of limitations in using encoders and normalizing flows (and aiming to solve class-conditional density estimation) as an alternative to typical classification networks. This would benefit quite a bit from ablations as well as discussion of these limitations, e.g., how well normalizing flows work here, choices that matter most when others aim to reproduce the work, and the scaling with respect to the number of classes. + Section 4's loss function is uninteresting from a novelty perspective. As R1 states, it's exactly the ELBO with a uniform prior, and I'm surprised the authors did not make that connection. I'd strongly recommend they tone down the significance of the contribution here.